# Self-Perceived Changes in Physical Activity and the Relation to Life Satisfaction and Rated Physical Capacity in Swedish Adults during the COVID-19 Pandemic—A Cross Sectional Study

**DOI:** 10.3390/ijerph18020671

**Published:** 2021-01-14

**Authors:** Frida Eek, Caroline Larsson, Anita Wisén, Eva Ekvall Hansson

**Affiliations:** Division of Physiotherapy, Department of Health Sciences, Lund University, 22240 Lund, Sweden; Caroline.larsson@med.lu.se (C.L.); anita.wisen@med.lu.se (A.W.); eva.ekvall_hansson@med.lu.se (E.E.H.)

**Keywords:** physical activity, life satisfaction, coronavirus pandemic, health

## Abstract

COVID-19 restrictions may prevent people from reaching recommended levels of physical activity (PA). This study examines self-perceived changes in the extent and intensity of PA during the COVID-19 pandemic, and the relation between perceived changes in PA and general life satisfaction and perceived physical capacity. A total of 1318 participants (mean age 47.8 SD12.6; 82.1% women) were recruited through social media in Sweden during autumn 2020. The survey included questions regarding perceived changes in PA compared to the previous year, the “Rating of Perceived Capacity” scale and “Life Satisfaction Questionnaire-11”. A change in PA was reported by 65% of participants. More participants reported an increase (36%) than a decrease (29%), however a decrease in PA was significantly more often considered to be due to the pandemic. The highest odds of decreased PA was found in the oldest age group (70+ years) (OR 2.8; 95% CI 1.4–5.7). Those who reported decreased levels of PA reported lower life satisfaction and aerobic capacity than the other groups (*p* > 0.001). Decreased physical activity was reported by many, but an equal share reported increased activity during the pandemic. The highest odds for decreased activity was found in the oldest group—the group that has been subjected to the strictest recommended COVID-19 restrictions in Sweden.

## 1. Introduction

To prevent the spread of COVID-19 and to enable health care systems to manage the increase in seriously ill persons, severe restrictions on daily life such as home confinement, quarantine, and social (physical) distancing have been implemented worldwide. Compared to strict lockdowns, the Swedish approach has not been as stringent as that in many other countries, but has instead relied on more voluntary responses. Still, measures to limit the spread of infection have been implemented in Sweden [1]. On-campus university teaching closed during spring 2020 and shifted to digital teaching and distance learning, the Swedish government has advocated working from home if possible, and fitness centers have restricted their activities. Restrictions regarding social (physical) distancing and gatherings of people have also been applied according to recommendations from the Swedish public health agency [2].

It was highlighted at the beginning of the pandemic that keeping up one’s physical activity during times of social restrictions and lockdowns is an important strategy for maintaining both physical and mental health in the population [3,4]. However, emerging evidence suggests declines in physical activity (PA) levels during the pandemic. Preliminary results from an international study showed a negative effect of home confinement on PA behavior, with a 33% decrease in the number of active minutes/day and a significant increase in sitting time (from 5 to 8 h per day), indicating a more inactive lifestyle during the pandemic [5]. Data from Fitbit, Inc., who collected physical activity data from 30 million users of wearable activity tracking devices during a week in March 2020, showed a reduction of 7–38% in average step counts in most countries compared with the same period the preceding year. In Sweden, the decline in the average step count was 9% [6]. A study from Belgium found that both self-reported exercise and sitting time had increased in a sample with a majority of highly active participants. The main reported obstacle for exercise among participants both with high and low activity levels before the pandemic was closed infrastructure. For people who were highly active already before the pandemic, cancelled sport events and activities and the absence of friends to exercise with were frequently reported obstacles. For participants with low activity levels, the fear of contamination with COVID-19 was also among the most frequently reported obstacles [7]. 

A Canadian study showed differences between previously active and inactive persons regarding change in physical activity behavior during the pandemic. A larger share of previously inactive persons had decreased their activity during the pandemic compared with previously active persons [8]. However, a considerable share also increased their activity, both among previously active and inactive persons. Moreover, increased activity was associated with better mental health and wellbeing in the inactive population. A Swedish study showed that 72% of female and 40% of male elite athletes who responded to a survey reported that they were feeling psychologically worse during the pandemic [9]. However, the perceived impact on patterns of physical activity in the more general Swedish population has not yet been examined. Increased knowledge of how COVID-19 may have affected the Swedish population’s ability to participate in physical activity and which groups seem most inhibited or encouraged to maintain PA will enable a strengthened ability to limit the negative health effects of COVID-19.

The aim of this study was therefore to describe self-reported physical activity levels and perceived changes in the extent and intensity of physical activity as well as physical capacities during the COVID-19 pandemic. A further aim was to explore if the perceived changes (increased, decreased, or no change) in physical activity differ between groups based on gender, age, education, family situation, geographical area, place of residence, and country of birth. Finally, we aimed to compare the perceived physical capacity and general life satisfaction between groups reporting increased, decreased, or no change in physical activity.

## 2. Materials and Methods 

### 2.1. Recruitment of Participants and Data Collection

Participant recruitment took place from September 1st 2020 to October 7th 2020, with information/advertising on social media. Participants had to be 18+ years old and able to read and understand Swedish. A Facebook page was created with information about the project, and advertisements were directed to parts of Sweden with large outbreaks (Stockholm and Gothenburg) and to Skåne, the southern region of Sweden, which consists of rural areas, smaller towns, and medium-sized cities. The link to the invitation was also posted on other forums such as Instagram and Twitter. The project webpage (hosted at Lund University.se) contained general project information, a participant information sheet, and a link to the survey. Participants gave their consent to participate by clicking on the link that directed them to the survey. 

### 2.2. Instrument (Survey)

The survey was created and managed using Research Electronic Data Capture (REDCap) [10,11], an electronic data capture tool designed to support data capture for research studies, hosted at Lund University.

The questionnaire included questions regarding basic demographic data, self-reported physical activity and physical capacity, and perceived changes in physical activity and capacity. 

Demographic information regarding age, gender, education (dichotomized into maximum secondary or post-secondary education), relationship status (single, married/cohabiting, or in a relationship but not living together), family situation (with or without children younger than 18 living at home either all the time or intermittently), information on geographical location (Stockholm (capital), Gothenburg (second largest city), Skåne (area around third largest city), or “elsewhere” in Sweden), residential community size (major city, middle-sized town, or small village), and country of birth (Sweden or other) was collected. Participants also responded to whether they had any current disease (requiring medication or treatment).

Current self-reported physical activity was measured with the International Physical Activity Questionnaire Short Form (IPAQ-SF) [12]. The IPAQ records the frequency and duration of physical activity at four intensity levels—(1) vigorous intensity activity (e.g., running, heavy lifts), (2) moderate intensity activity (e.g., leisure cycling, swimming), (3) walking, and (4) sitting—during the last seven days. 

Participants were also asked whether they have a smart phone or any other activity-monitoring tool. If they had, they were asked to report their recorded average steps/day during the last month.

Participants rated their current perceived aerobic capacity on the Rating of Perceived Capacity scale (RPC) [13]. The RPC scale was developed in Sweden and is based on metabolic equivalents (METs) from 1 to 20 on a progressive scale and linked to physical activities. Participants were asked to choose the most strenuous activity, with corresponding MET value, that could be sustained for at least 30 min. According to a validation study performed in Norway, the RPC scale might overestimate VO2max at the individual level but is considered a valuable tool for the estimation of aerobic capacity in large-scale studies [14].

Life satisfaction was measured by the Life Satisfaction Questionnaire-11 (LISAT-11) [15,16]. LISAT-11 includes one global item for “life as a whole” and 10 domain-specific items for “vocational situation”, “financial situation”, “leisure”, “contact with friends”, “sexual life”, “activities of daily living”, “family life”, “partnership/relationship”, “physical health”, and “psychological health”. Items are rated on an ordinal scale ranging from 1 (very dissatisfying) to 6 (very satisfying). The mean score for the 11 items was computed (mean total LISAT score).

Perceived changes in physical activity were measured by six items asking the participants to report whether they had changed (increased, decreased, or no change) their current extent and/or intensity of physical activity on a moderate or vigorous intensity level and changed their extent of walking and sitting compared with same time the previous year. If reporting a change, the participants were further asked to indicate whether they considered the change to be due to the pandemic and its entailed restrictions (yes, partly, or no). Participants were also asked to indicate whether they perceived that their aerobic capacity, muscular strength, and balance had changed (increased, decreased, or no change) compared with the previous year and, if it had changed, whether or not they considered the change to be due to the pandemic and its entailed restrictions. 

### 2.3. Statistics

#### 2.3.1. Data Management

Physical activity according to IPAQ was transformed into metabolic equivalent (MET) minutes according to the short-form scoring protocol [17]. Minutes of activities per week were computed as minutes per session x frequency of session per week and transformed into metabolic equivalent (MET) minutes by multiplying with the respective constant for walking (3.3), moderate intensity activity (4.0), and vigorous intensity activity (8.0). The total physical activity MET minutes/week was computed as the sum of walking, moderate, and vigorous MET minutes/week and categorized into low, moderate, and high intensity, as defined in the short-form scoring protocol.

The four questions regarding change in (1) extent and (2) intensity within physical activity on (a) a vigorous and (b) moderate intensity level were coded as -1 (decrease), 0 (no change), or +1 (increase) and summarized into a score on a scale ranging from −4 (decrease in all aspects) to +4 (increase in all aspects). Negative values (-1–−4) were categorized as “decrease” in PA, 0 was categorized as “no change”, and positive numbers (1–4) were categorized as an “increase” in PA. 

The change variable was also dichotomized into two variables: “Increased activity” (increase vs. no change or decrease) and “Decreased activity” (decrease vs. no change or increase).

For perceived reasons for changes, the responses “yes” or “partly” to the question “do you consider the change to be due to the COVID pandemic and entailed restrictions” were categorized as “due to pandemic with entailed restrictions”.

#### 2.3.2. Analysis

Due to the skewed distribution of the IPAQ data, current physical activity levels (METS and average step count) are presented as medians and inter quartile ranges (Q1–Q3). 

Perceived changes are presented as the proportion (%) of participants reporting an increase, decrease, or no change in the different aspects of physical activity, behaviors, and capacity. Chi square tests were used to compare the proportion of participants who considered the change to be due to the pandemic and its entailed restrictions, between those who reported an increase/improvement and those who reported a decrease/deterioration.

The outcome variable in the analyses regarding change in PA is the combined “perceived change in physical activity” (combining extent and/or intensity), which is categorized into increase, no change, or decrease. Further outcomes are life satisfaction (mean total LISAT score) and rated perceived capacity (RPC). Independent/exposure variables were gender, age, education, civil status, family situation, geographical area, residential community size, and country of birth. 

Chi square tests were performed in order to explore if the proportion of persons reporting a decrease, increase, or no change in activity differed between groups based on gender, age, education, civil status, family situation, geographical area, place/type of residence, and country of birth. All the variables for which the chi square test indicated that the proportions differed significantly between groups were entered into a multivariable logistic regression analysis in order to examine the adjusted odds ratios (OR) for the two dichotomized outcomes (increased and decreased activity) in each group. Variables where the change in activity did not differ significantly between the groups according to the chi square test were not considered to have a confounding effect on the results and were hence not included in the multivariable model. However, the variable “current disease” was evaluated for its potential confounding effect according to the “change in estimate” approach [18] by adding the variable to the model and including it if the inclusion resulted in a change in estimate (OR) of >15%. The results are presented as odds ratios (OR) with corresponding 95% confidence intervals (95% CI). 

Comparisons regarding life satisfaction (LISAT) and perceived physical capacity (RPC) between groups reporting a decrease, increase, or no change in physical activity were performed through an analysis of variance (ANOVA), with pairwise post hoc comparisons in the case of a significant omnibus test. Age, residential community size, and “current disease” were introduced in the model and evaluated for their potential confounding effect according to a change in estimate approach (>15% change in group differences). The significance level was set to *p* < 0.05. All the analyses were performed in Statistical Package for Social Sciences (SPSS) (IBM SPSS, Version 26.0. USA, Armonk, NY: IBM Corp).

#### 2.3.3. Ethics

The study was performed according to the Helsinki Declaration and approved by the Ethical Review Authority in Sweden, drn 2020-02776.

## 3. Results

A total of 1318 persons answered the questionnaires. A description of the sample is presented in Table 1.

The median current level of total (walking, moderate intensity activity, and vigorous intensity activity) physical activity in the group was 2902 (Q1–Q3: 1746–4384) METS/week. The median level of vigorous intensity was 1200 (400–1920), that for moderate activity was 720 (240–1440), and that for walking was 693 (347–1386) METS/week. Among those who responded to IPAQ (*n* = 1144), 64% fulfilled the criteria for high activity level, 29% were moderately active, and 7% had low activity. Almost half of the participants (*n* = 630; 48%) had recorded step counts from some activity tracker. The median reported average daily steps during the last month was 10,000 (Q1–Q3: 7305–12,000).

Approximately half of the participants reported a perceived change in physical activity behaviors (extent and/or intensity), with a relatively equal distribution between reported increase and decrease in vigorous physical activity but with a slightly larger share reporting an increase than a decrease in the extent and intensity of moderate physical activity and walking (Table 2). A larger share also reported an increase rather than a decrease in the extent of time spent sitting. When combined, 36% of the participants reported an increase, and 29.8% reported a decrease in the extent and/or intensity of moderate and/or vigorous activity (Table 2). A slightly larger share reported a perceived increase rather than a decrease in aerobic capacity.

The majority of participants perceived the changes—both increase and decrease in physical activity and capacity—to be due to the pandemic and its entailed restrictions. However, a significantly larger share among those reporting a decrease in activity, increase in sitting/sedentary time, or deterioration of physical capacity considered the change to be due to the pandemic and its entailed restrictions compared to the participants reporting increased activity or improved capacity (*p* ≤ 0.003) (Table 2).

The proportions of reported change (increase, decrease, or no change) in physical activity of moderate or vigorous intensity differed significantly between age groups and residential community size (Table 3). The odds for decreased activity were significantly higher in the oldest age group (70+) compared to the youngest (OR 2.8, 95% CI 1.4–5.7). The odds for increased activity were significantly lower in both age groups 70+ and 50–69 compared with the youngest age group. Persons living in big cities had significantly higher odds (OR 1.5 (95% CI 1.1–2.0) of increased activity compared to persons living in a small village (Table 4). There were no significant differences regarding perceived change in physical activity between groups based on age, family situation, geographical area, or country of birth (Table 3). 

Participants reporting decreased activity reported significantly lower aerobic capacity (RPC) and life satisfaction (mean total LISAT score) compared with persons reporting no change or increased activity level (*p* < 0.001) (Figure 1 and Figure 2).

## 4. Discussion

A majority of the participants reported a change in physical activity and/or capacity during the COVID-19 pandemic in comparison with the same time the preceding year. Approximately 20–30% reported a decrease in the different aspects of physical activity. Equally large, or even larger shares, however, reported increased physical activity, and improved physical capacity. Participants who reported a decrease or deterioration in physical activity or function more frequently attributed the change to the pandemic and its entailed restrictions than those reporting an increase or improvement.

The group that had decreased their overall physical activity reported significantly lower life satisfaction and lower current aerobic capacity compared with the groups that reported increased or unchanged physical activity during the pandemic. The highest odds for decreased activity were found in the oldest age group (70+ years), which is the group that in Sweden has been subject to specifically stricter recommended restrictions. Persons living in bigger cities had higher odds of increased activity compared with persons living in small villages.

The current activity level in the study sample was relatively high compared with what has been shown in a previous population-based study of the general Swedish population [19]. The majority of the participants in our study were classified as having a high activity level. Physical inactivity contributes to a wide range of unhealthy conditions, while physical activity has significant health benefits, aids the prevention of non-communicable diseases, and is associated with reduced mortality [20,21,22,23]. Exercise can also be used as potential treatment or therapy in many chronic diseases [24]. In order to achieve such health effects, adults are recommended to be “moderately physically active (i.e., a moderate amount of effort that noticeably accelerates the heart rate) at least five days a week for a minimum of 30 min a day”—i.e., at least 150 min per week—or engage in vigorously intense activity for at least 75 min per week. The aerobic activity should preferably be combined with muscle strengthening activities at least two days per week [25]. For additional health benefits, at least 300 min at moderate intensity or 150 min at vigorous intensity is recommended. However, those thresholds have been recently challenged, as the focus on a “minimal level” may induce unnecessary barriers to those who would benefit by just increasing their activity, even if below the threshold [21,26]. This is also highlighted in the newly updated recommendations from the WHO, which states that any amount of PA is better than none, but more is better [27]. 

While studies from other parts of the world have indicated decreased physical activity levels during the pandemic [5,8], including preliminary reports of fit bit users in Sweden [6], we could not confirm this pattern in our study. A considerable share reported a decrease in activity, but an equal or even larger share of participants reported that they had instead increased their activity compared with same period during the previous year. A potential reason for this could be the relatively mild restrictions that have been applied in Sweden. The recommendations regarding physical distance have affected the availability of group training activities, and remote work has reduced physically active commuting. However, gyms have still been open and reduced time for commuting may provide extended time for alternative forms of physical activity.

Persons living in major cities had higher odds of increased activity compared with persons living in small towns. This correspond to previous studies that have shown that area-level factors may influence physical activities such as walking/bicycling and that high area density is associated with more activity even after controlling for socioeconomic factors [28]. However, we found no differences between the different geographical regions in Sweden, even though there were clear differences regarding the outbreak of infections where, for example, at the time of the data collection the capital Stockholm had a considerably larger spread of COVID-19 infection compared with the southern district (Skåne).

Although approximately equal proportions reported an increase compared to a decrease in physical activity and capacity, the reported decreases were to a greater extent perceived to be due to the pandemic and its restrictions. This indicates that the pandemic is perceived to have negative effects on the activity level. Previously, an impact on individuals’ daily activities and mental health with higher levels of stress and anxiety were shown among those who experienced a decreased amount of physical activity [29]. In this study, the participants reporting decreased physical activity also reported both significantly lower levels of life satisfaction and a significantly lower rated aerobic capacity. The results were not confounded by factors such as age or current general health status, but due to the cross-sectional design we cannot draw any conclusions regarding the direction of causality. Lower life satisfaction could be both a cause and an effect of decreased activity. Our results are, however, in line with the study by Lesser et al. [8] that found better mental health and wellbeing among inactive persons who had increased their activity during the pandemic. In our study, the potential effect was, however, more reflected in the lower life satisfaction among persons who had decreased their activity. 

Since both decreased physical activity and lower aerobic capacity are independent risk factors for non-communicable diseases and associated mortality [20,21,22,23], these findings indicate the challenge in further highlighting the recommendations to both increase physical activity levels and stimulate aerobic training. Since other forms of physical activity, such as strength training, also have shown positive effects on both mental and physical health [30,31,32,33], all forms of physical activity should be encouraged and promoted.

The group that seems to be most affected in terms of decreased activity was the group that has also been subjected to the strictest recommended restrictions in Sweden—i.e., adults aged 70+ years. This is a group that seems more prone to long-term negative health-related consequences through reduced physical activity. Even short periods of step reduction (<2 weeks) in healthy older adults have previously been shown to have detrimental effects of reductions in muscle mass and impairments in insulin sensitivity and systemic inflammatory markers [34]. While younger individuals recover after a short period of inactivity, even more intense training protocols do not seem to rebuild the lost muscle mass and reestablish glucose metabolism in older individuals [35,36]. It is a delicate balance though, as there are reasons for the stricter recommended restrictions for this age group. Older persons are subject to a higher risk for severe infections and a higher mortality from COVID-19 [37]. Those harsher restrictions seem to have inhibited maintained physical activity in this age group to a higher extent than in younger age groups that had relatively milder recommended restrictions during the study period. For this older age group though, visiting gyms for example may be associated with a risk of infection that could outweigh the potential benefit. Instead, focus should be on encouraging and finding ways to maintain or increase physical activity in forms that are possible to perform while following recommended restrictions regarding, e.g., physical distance. Outdoor exercise or activity and/or home-based exercise programs or online programs that can be followed and performed from home are examples of activities that can be performed with safe physical distances. The COVID-19 pandemic will continue to demand extensive adaptations for the global society. Although the implemented restrictions are necessary for society at large to prevent the health care system from becoming overwhelmed, reduced physical activity, especially among older adults, can have major negative consequences for individuals’ health, quality of life, morbidity, and mortality. Physical activity may also have a protective effect for severe COVID-19 [38]; thus, efforts to maintain or even increase levels of physical activity during the pandemic might contribute to less severe health effects. Again, the milder restrictions applied in Sweden, such as permissibility to move around outdoors, which has led to more people moving around in parks and green areas, might contribute to this seemingly mild reduction in PA levels in our sample. As we are now facing the second wave of the pandemic with increased restrictions, it is crucial to have an amplified focus on promoting and enabling people to engage in physical activity and reducing sedentary behavior, in particular among older adults, to limit the far-reaching impact and unknown long-term health implications of COVID-19.

The strength of this study was the relatively large sample size and that participants from all age groups answered the questionnaire. The participants came from different environments, such as small villages, medium-sized towns, larger cities, and different geographical areas. Thus, it may be possible to generalize our results to some extent, however with caution. Women and persons with higher education seem to be over-represented in our sample. This may likely be due to a selection bias from the recruitment via social media. The current activity levels further indicate that the sample includes relatively highly active people. This may likely be due to non-response bias leading to an over-representation of persons with an interest in physical activity. This may limit the generalizability to the general Swedish adult population, and potentially over-estimate the positive changes in physical activity during the pandemic. However, it is less likely that the between-group differences that we found within the sample are affected by this bias. 

Another limitation is that the results are based on self-reported perceptions of change in activity behavior rather than objectively measured changes. The questions regarding changes in physical activity were formulated specifically for the current study and have hence not been tested for validity or reliability. Since we did not collect any pre-pandemic measures, the alternative would have been to ask for retrospective reports of physical activity levels before the pandemic. Although IPAQ is a widely used instrument and considered one of the best tools to measure self-reported physical activity, the validity of those measures is still limited. The validity of retrospective reports of physical activity or RPC is not known, but would probably be lower than reports of current levels. We hence considered the best alternative to be to ask directly about perceived changes in the different aspects of physical activity. By using study-specific questions, we could also cover perceived changes in different aspects of physical capacity. The measure of current activity was also based on self-reports but from an established instrument (IPAQ). As previously mentioned, the validity of self-reported measures of physical activity is not optimal. Objective measures of physical activity such as using accelerometers might have provided a more accurate measure of current status [39]. Approximately half of the participants, however, reported results regarding current activity level based on data from private activity tracers, confirming the finding that the sample had a relatively high activity level.

## 5. Conclusions

This study shows that while a considerable share of the participants reported decreased activity during the pandemic, approximately the same amount of participants reported increased activity during the same period. Persons reporting decreased activity had significantly lower life satisfaction and rated their perceived aerobic capacity lower compared to the groups with increased or unchanged activity. Although a significantly larger share of the participants that had decreased their activity, compared with those who had increased their activity, considered the change in activity was attributed to the pandemic and its entailed restrictions, the relatively milder restrictions applied in Sweden may have contributed to a better maintained physical activity level in the population. The group with the highest odds of decreased activity was the 70+-year-olds—the group that has been subject to the strictest recommended restrictions in Sweden.

## Figures and Tables

**Figure 1 ijerph-18-00671-f001:**
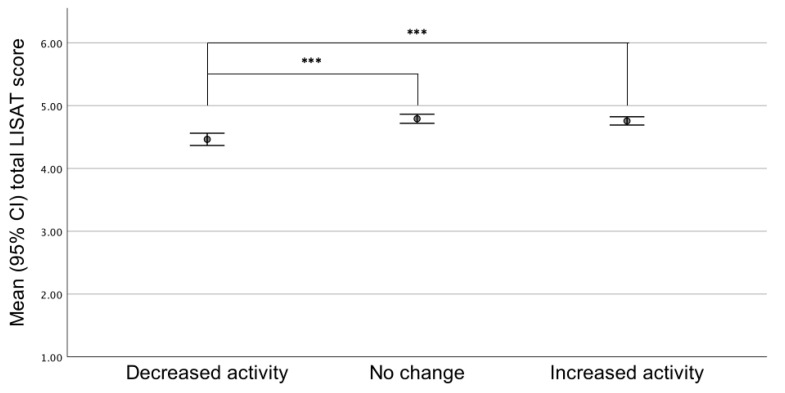
Life satisfaction (LISAT) score in groups reporting decrease, no change, or increase in activity level (intensity and extent); *** *p* < 0.001.

**Figure 2 ijerph-18-00671-f002:**
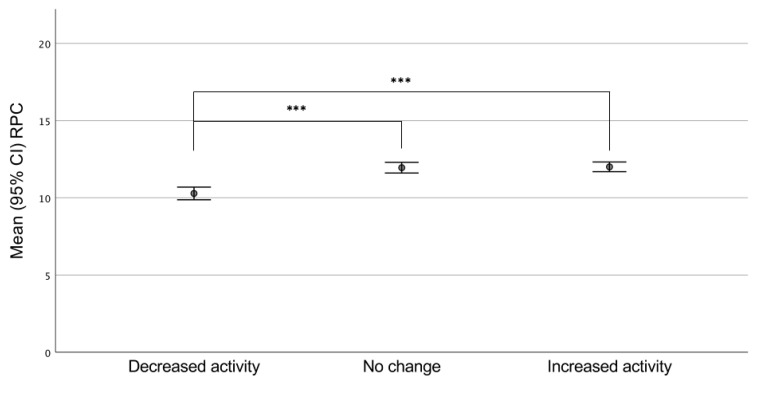
Rated physical capacity (RPC) in groups reporting decrease, no change, or increase in activity level (intensity and extent); *** *p* < 0.001.

**Table 1 ijerph-18-00671-t001:** Description of study sample (*n* = 1318).

Variable	
Gender % (*n*)	
Female	82.1 (1060)
Male	17.4 (225)
Other/do not want to tell	0.5 (5)
Age mean (SD)	47.8 (12.6)
Age group % (*n*)	
<35	15.8 (200)
35–49	40.2 (508)
50–69	39.2 (496)
70+	4.8 (61)
Education % (*n*)	
Secondary school	16.5 (218)
Post-secondary education	81.3 (1071)
Civil status % (*n*)	
Single	18.9 (244)
Married/cohabiting	74.2 (957)
Partner, not cohabiting	6.9 (89)
Family situation % (*n*)	
Children living at home full time	38.4 (495)
Children living at home part time	5.7 (74)
No children living at home	55.9 (720)
Geographical area % (*n*)	
Stockholm	21.9 (282)
Gothenburg	14.4 (186)
Skåne	42.9 (553)
Other parts	20.9 (269)
Residential community size % (*n*)	
Small village	30.0 (387)
Middle size town	32.8 (423)
Big city	37.1 (478)
Country of birth % (*n*)	
Sweden	90.7 (88.5)
Other	9.3 (120)
RPC Mean (SD) (*n* = 1102)	11.5 (3.4)
LISAT score mean (SD) (*n* = 1082)	4.7 (0.7)

RPC = rated perceived capacity; LISAT = Life Satisfaction Questionnaire-11.

**Table 2 ijerph-18-00671-t002:** Reported change in physical activity behaviors and capacity compared with the same time the previous year, and perceived change due to COVID-19-related restrictions.

Variables	% (*n*)	If Change: Considered Due to COVID-19 Related Restrictions % (*n*)	*p* *
**Extent of vigorous activity**			
+	28.2 (288)	69.4 (200)	
0	45.0 (460)		
-	26.9 (275)	88.0 (241)	<0.001
**Intensity of vigorous activity**			
+	21.5 (219)	60.3 (132)	
0	54.7 (557)		
-	23.8 (242)	85.1 (205)	<0.001
**Forms of vigorous activity**			
Same	45.8 (468)		
Different	34.0 (347)		
No forms	20.2 (206)		
**Extent of moderate activity**			
+	29.1 (202)	75.4 (224)	
0	52.5 (689)		
-	18.4 (131)	88.8 (167)	<0.001
**Intensity of moderate activity**			
+	19.8 (202)	75.2 (152)	
0	67.4 (689)		
-	12.8 (131)	88.5 (116)	0.003
**Forms of moderate activity**			
Same	59.8 (610)		
Different	35.2 (359)		
No forms	5.0 (51)		
**Summarized/combined change ****			
+	36.0 (364)	71.4 (260)	
0	34.2 (346)		
-	29.8 (301)	89.0 (268)	<0.001
**Extent of walking**			
+	39.2 (401)	79.3 (318)	
0	46.7 (477)		
-	14.1(144)	65.3 (94)	0.001
**Extent of sitting**			
+	29.6(23.0)	89.4 (269)	
0	57.8(44.9)		
-	12.6 (9.8)	59.4 (76)	<0.001
**Aerobic capacity**			
+	31.9 (326)	61.0 (192)	
0	46.7 (477)		
-	21.4 (219)	80.9 (169)	<0.001
**Strength**			
+	26.0 (266)	53.0 (140)	
0	50.2 (513)		
-	23.8 (243)	81.8 (198)	<0.001
**Balance**			
+	11.5 (117)	48.7 (57)	
0	81.5 (832)		
-	7.1 (72)	76.4 (55)	<0.001

* Chi square test comparison of the proportion considering the change to be due to the pandemic and its related restrictions in the groups reporting increased or decreased activity; ** change in extent or intensity of moderate or vigorous activity.

**Table 3 ijerph-18-00671-t003:** Proportion of participants reporting decreased, increased, or unchanged physical activity in different groups based on gender, age, family situation, geographical area, type of village, and country of birth.

	Decreased Activity *	Unchanged Activity *	Increased Activity *	*p* **
	**% (*n*)**	**% (*n*)**	**% (*n*)**	
**Gender**				
Women	29.9 (245)	34.2 (280)	35.9 (294)	
Men	27.2 (50)	35.9 (66)	37.0 (68)	0.758
**Age group**				
<35	30.1 (43)	21.7 (31)	48.3 (69)	
35–49	25.6 (102)	34.1 (136)	40.4 (161)	
50–69	30.9 (123)	38.4 (153)	30.7 (122)	
70+	55.8 (24)	27.9 (12)	16.3 (7)	**<0.001**
**Education**				
Secondary school	28.1 (43)	36.6 (56)	35.3 (54)	
Post-secondary education	29.9 (298)	34.0 (346)	36.2 (363)	0.808
**Civil status**				
Single	35.1 (67)	34.6 (66)	30.4 (58)	
Married/cohabiting	28.2 (211)	34.7 (259)	37.1 (277)	
Partner, not cohabiting	30.4 (21)	30.4 (21)	39.1 (27)	0.303
**Family situation**				
Children living at home full time	24.8 (98)	35.9 (142)	39.2 (155)	
Children living at home part time	30.0 (18)	38.3 (23)	31.7 (19)	
No children living at home	33.2 (183)	32.6 (180)	34.2 (189)	0.078
**Geographical area**				
Stockholm	27.8 (62)	30.0 (67)	42.2 (94)	
Gothenburg	27.8 (42)	30.5 (46)	41.7 (63)	
Skåne	29.5 (130)	36.6 (161)	33.9 (149)	
Other part	33.7 (65)	36.8 (71)	29.5 (57)	0.093
**Residential community size**				
Small village	27.8 (87)	39.9 (125)	32.3 (101)	
Middle size town	31.9 (100)	34.5 (108)	33.5 (105)	
Big city	29.0 (110)	29.6 (112)	41.4 (157)	**0.025**
**Country of birth**				
Sweden	29.4 (270)	35.1 (322)	35.5 (326)	
Other	33.0 (29)	27.3 (24)	39.8 (35)	0.338

* Change in extent or intensity of moderate or vigorous activity. ** Chi square test, significant *p*-values (<0.05) presented in bold.

**Table 4 ijerph-18-00671-t004:** Odds ratios (OR) for increased and decreased activity in different age groups and place of residence.

	**Decreased Activity *** **OR (95% CI)**	**Increased Activity *** **OR (95% CI)**
**Age group**		
<35	1 (ref)	1 (ref)
35–49	0.8 (0.5–1.3)	0.7 (0.5–1.1)
50–69	1.0 (0.7–1.6)	**0.5 (0.3–0.7)**
70+	**2.8 (1.4–5.7)**	**0.2 (0.1–0.5)**
**Residential community size**		
Small village	1 (ref)	1 (ref)
Middle-sized town	1.2 (0.8–1.7)	1.0 (0.7–1.5)
Big city	1.1 (0.8–1.5)	**1.5 (1.1–2.0)**

* Change in extent and/or intensity of moderate and/or vigorous activity. Abbreviations: OR = Odds Ratio; CI = confidence interval. Significant results are presented in bold.

## Data Availability

Data is available only on request to the authors, according to the ethical approval from the Swedish Ethical Authority.

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
