# Peer review of "Self-Perceived Changes in Physical Activity and the Relation to Life Satisfaction and Rated Physical Capacity in Swedish Adults during the COVID-19 Pandemic—A Cross Sectional Study"

_ijerph, 2021, doi:10.3390/ijerph18020671_

Round 1

Reviewer 1 Report

Dear authors,

Thank you for the opportunity to revise your work entitled: Self-perceived changes in physical activity in Sweden during the Covid-19 pandemic –a cross sectional study. After analyzing the paper, I do consider that the methods sections could be improved and that some control missing in some of the analysis are a major limitation. Detailed comments to the paper are as follows.

>> Title

It should be reviewed. Does not include nothing to general life satisfaction or perceived physical capacity. The population could be specified Swedish adults and older adults instead of “Sweden”.

>> Abstract

When defining the purposes it is stated that “study examines self-perceived changes in PA levels, aerobic capacity and behaviors”. Consider reformulating this sentence. What are the other behaviors you are mentioning here? Sedentary behaviour? Other? Which beahviours?

>> Introduction

Why is this study important? This must be made much clearer in the introduction.

State the prespecified hypotheses

>> Methods

Was the Rating Perceived Capacity Scale already validated for the Swedish Context.

Page 3, L120-128. What is the reliability/validity of the six items asking about the perceived PA changes? Some other validated instruments /question could have been used to report previous activity in the past 12 months. Why the authors did not chose to use some of them? As this is a central variable in the study, clarifications are important on this.

Clearly define all outcomes, exposures, predictors, potential confounders, and effect modifiers.

Further describe any efforts to address potential sources of bias.

“Variables where the chi square test indicated that the proportions differed significantly between groups were entered into a multivariable logistic regression analysis in order to examine the odds ratios (OR) for the two dichotomized outcomes (Increased- and Decreased activity) in each group”. Why was this criterion chosen? Please further justify.  This decision seems to limit the opportunity to control for confounding factors in the subsequent analysis (Table 4).

In addition, I do consider to be critical the lack of several controlling factors, scuh as, education, socioeconomic status, sex, BMI, health condition (it might be different for those aged <35yrs and >70 yrs); geographical area, etc. In my opinion, the authors controlling the analysis for some important controlling factors seems to be missing. Again for data presented in figure 1 why use ANOVA and not ANCOVA? This is a critical point regarding the study. In my view, should be reconsidered.

>> Discussion

Further discuss the generalisability (external validity) of the study results.

Sincerely.

Author Response

We want to thank the reviewers for a thorough review with valuable comments and input, that helped us to further clarify and improve the manuscript. Please see our detailed point-by-point responses and actions below.

Reviewer 1

Reviewer 1 comment: >> Title: It should be reviewed. Does not include nothing to general life satisfaction or perceived physical capacity. The population could be specified Swedish adults and older adults instead of “Sweden”.

Author’s action: Thank you for pointing out this lacking information. We have changed the title to “Self-perceived changes in physical activity and the relation to life satisfaction and rated physical capacity in Swedish adults during the Covid-19 pandemic –a cross sectional study”

Reviewer 1 comment: >> Abstract: When defining the purposes it is stated that “study examines self-perceived changes in PA levels, aerobic capacity and behaviors”. Consider reformulating this sentence. What are the other behaviors you are mentioning here? Sedentary behaviour? Other? Which beahviours?

Author’s response: We initially meant the general behavior (including extent and intensity), as well as the question regarding change in “forms” of activity. We however agree that this may not be clear to the reader.
Author’s action:
Due to the word limit, there is no space for further specification of “behavior” so we reformulated the aim in abstract to “This study examines self-perceived changes in extent and intensity of PA during the Covid-19 pandemic” (row 12)

Reviewer 1 comment: >> Introduction: Why is this study important? This must be made much clearer in the introduction.

Author’s response: The purpose of the study is to contribute with increased knowledge about how Covid-19 may have affected the Swedish population's ability to participate in physical activity, and which groups that seem to have been most inhibited, or encouraged, to maintain PA levels during the pandemic.  This knowledge may enable a strengthened focus to limit negative health effects among these.
Author’s action: We have now added following statement (page 1 row 63-66) “An increased knowledge on how Covid-19 may have affected the Swedish population's ability to participate in physical activity, and which groups that seem most inhibited or encouraged to maintain PA will enable a strengthened focus to limit negative health effects of Covid-19.

Reviewer 1 comment: State the prespecified hypotheses

 Author’s response: Since this was an exploratory study, we did not have any pre-specified hypotheses (in the sense that we assume the reviewer means; I e one-sided/directed hypotheses). We expected worse life satisfaction and RPC among those who had reduced their activity level, but with the overall exploratory approach in the study we would prefer not to spell out directed hypotheses, and we consider the general two sided hypotheses ( “any difference”) to be included in the research questions.

Reviewer 1 comment:>> Methods: Was the Rating Perceived Capacity Scale already validated for the Swedish Context.

Author’s response: The RPC scale was developed in Sweden (Wisén et al). The scale has also been validated, however this validation study was performed in Norway.
Author’s action: We have now added this information (page 3 row 116) “Participants rated their current perceived aerobic capacity on the Rating of Perceived Capacity scale (RPC) [21]. The RPC scale was developed in Sweden and…..” and (row 119-21) “According to a validation study performed in Norway, the RPC scale might overestimate VO2max at individual level but is considered a valuable tool for estimation of aerobic capacity in large scaled studies”

Reviewer 1 comment: Page 3, L120-128. What is the reliability/validity of the six items asking about the perceived PA changes? Some other validated instruments /question could have been used to report previous activity in the past 12 months. Why the authors did not chose to use some of them? As this is a central variable in the study, clarifications are important on this.

Author’s response: The questions regarding change was formulated specifically for the current study, and hence not tested for validity or reliability. The alternative would have been to apply a retrospective report of IPAQ. However, although IPAQ is widely used and often considered the best available tool for measure of self reported physical activity, the validity is yet low, and many responders usually misunderstand instructions and hence provide unrealistic responses. In the report of current activity level, we handled those issues according to suggestions in the manual. However, IPAQ (or other potential measures) is not validated for retrospective report and based on the questionable validity even of current report, we decided to rather focus on the direct question of (perceived) changes. This way, we could also cover different aspects of change in both activity and capacities.

Author’s action: We have now included a discussion regarding this limitation, in the methods discussion (page 12, row 366-76). “The questions regarding change in physical activity was formulated specifically for the current study, and has hence not been tested for validity or reliability. Since we did not have any pre-pandemic measures collected, the alternative would have been to ask for retrospective report of physical activity levels before the pandemic. Although IPAQ is a widely used instrument, and considered one of the best tools to measure self reported physical activity, the validity of those measures is still limited. The validity for retrospective reports of physical activity or RPC is not known, but would probably be lower than reports of current levels. We hence considered the best alternative to be to ask directly about perceived changes in the different aspects of physical activity. By using study specific questions, we could also cover perceived changes in different aspects of physical capacity.  The measure of current activity was also based on self-reports, but from established instrument (IPAQ). As previously mentioned, the validity for self reported measures of physical activity is not optimal.”

Reviewer 1 comment: Clearly define all outcomes, exposures, predictors, potential confounders, and effect modifiers.

Author’s response: Thank you for noticing that this information was incomplete.

Author’s action: We have clarified the different included variables in the statistics section (page 4, 165-69 and185-87). “The outcome variable in the analyses regarding change in PA is the combined “perceived change in physical activity” (combining extent and/or intensity) categorized into increase, no change, or decrease. Further outcomes are life satisfaction (mean total LISAT score) and rated perceived capacity (RPC). Independent/exposure variables were gender, age, education, civil status, family situation, geographical area, residential community size and country of birth………… the variable “current disease” was evaluated for potential confounding effect….”

Reviewer 1 comment: Further describe any efforts to address potential sources of bias.

Author’s response: The main source of bias in this study is the selection- and non response bias. We directed the invitation as broad as possible and encouraged participation regardless of current activity level. The advertisement/invitation was directed to participants from different environments such as small villages, medium sized towns as well as larger cities and different geographical areas. We also directed the study parts of Sweden with large outbreaks respectively smaller outbreaks. Regarding response bias and or validity in relation to the questionnaire, we used established and validated measures when available according to the research questions. An in-depth review of possible risks of bias is presented in the discussion section (page 12, row 359-78).

 Reviewer 1 comment: “Variables where the chi square test indicated that the proportions differed significantly between groups were entered into a multivariable logistic regression analysis in order to examine the odds ratios (OR) for the two dichotomized outcomes (Increased- and Decreased activity) in each group”. Why was this criterion chosen? Please further justify.  This decision seems to limit the opportunity to control for confounding factors in the subsequent analysis (Table 4).”

Author’s response: We are aware that there are different ways to handle potential confounding. According to Mickey and Greenland (1), the recommended way to evaluate confounding effects is the “change in estimate” approach. The reasons that we did not solely apply this approach are the following. The cross tabulation with accompanying chi square tests aims to give a presentation of the proportions of change in the different groups. Based on the definition of confounders to have an independent effect on (or at least association with) both exposure and outcome, we considered the variables where we found no difference in outcome (change) between the groups to not be considered confounding factors. Also, applying a backwards (or forward) stepwise inclusion by adding all independent/exposure variables in the multivariable model was not optimal, due to the “collinearity” (overlap) between the variables geographical location and residential community size. However, due to the reviewer’s concern, we have performed all models with multivariable adjustment (not simultaneous inclusion of geographical location and residential community size), but it did not change the results. According to reviewer’s suggestion, we also added educational level as independent variable, and included “current disease” (health status) for evaluation of potential confounding effects.

Author’s action: We have revised and clarified the description of statistical analyses and handling of potential confounding (page 4, row 172-76). “Variables where the change in activity did not differ significantly between the groups according to the chi square test, were not considered to have a confounding effect on the results and were hence not included in the multivariable model. However the variable “current disease” was evaluated for potential confounding effect according to “change in estimate” approach (ref Mickey and Greenland) by adding the variable to the model and including it if the inclusion result in a change in estimate ( OR) of >15%.”

Reviewer 1 comment: In addition, I do consider to be critical the lack of several controlling factors, scuh as, education, socioeconomic status, sex, BMI, health condition (it might be different for those aged <35yrs and >70 yrs); geographical area, etc. In my opinion, the authors controlling the analysis for some important controlling factors seems to be missing. Again for data presented in figure 1 why use ANOVA and not ANCOVA? This is a critical point regarding the study. In my view, should be reconsidered.

Author’s response: Thank you for pointing this put. We actually did perform an ANCOVA (ANOVA model adjusted for age), although we labeled it ANOVA. However, we have now evaluated potential confounding effects from all suggested variables except BMI, which we did not have information on. In this case, it is also unclear if BMI should be considered mainly a confounder, or a potential “link in the causal chain”. None of the potential confounders had any effect on the results (no change in estimate), but we have revised our description of the handling.

Author’s action: We have revised and clarified the description of statistical analyses and handling of potential confounding (page 4, row 178-83). “Comparisons regarding life satisfaction (LISAT) and perceived physical capacity (RPC) between groups reporting decrease, increase or no change in physical activity were performed through analysis of variance (ANOVA), with pairwise post hoc comparisons in case of significant omnibus test. Age, residential community size and “current disease” were introduced in the model and evaluated for potential confounding effect according to change in estimate approach (>15% change in group differences)”.

Reviewer 1 comment: >> Discussion: Further discuss the generalisability (external validity) of the study results.

Author’s response: We are aware that our sample may not be completely representative for the total adult Swedish population, due to some possible selection- and non-response bias. This may to some extent may the external validity/generalizability of our results. The general current physical activity may be over-estimated, just as the positive changes in physical activity during the pandemic. However, the within-group differences that we found should not be that affected by this potential bias.
Author’s action: We have added some further considerations regarding the generalizability to the methods discussion; page 12 row 355-65)
“The strength of this study was the relatively large sample size and that participants from all age groups answered the questionnaire. Participants came from different environment, such as small villages, medium sized towns as well as larger cities and different geographical areas. Thus, it may be possible to generalize our results to some extent, however with caution. Women, and persons with higher education seem to be over-represented in our sample. This may likely be due to a selection bias from the recruitment via social media. The current activity levels further indicate that the sample include relatively highly active people. This may likely be due to non-response bias leading to an over-representation of persons with interest in physical activity. This may limit the generalizability to the general Swedish adult population, and potentially over-estimate the positive changes in physical activity during the pandemic.  However, the within group differences that we found should not be that affected by this bias.

Reviewer 2 Report

The current study investigated subjective physical activity of citizens related to COVID-19 pandemic in Sweden. Sweden used rather unique strategy to fight against COVID-19 and its effects should be investigated thoroughly. The present study is one of these describing the reality of what has happened in Sweden during pandemic. Therefore, the current manuscript has its own originality and value to be published.

However, I would like to suggest several points to improve the article.

1. The abstract should be better informative.

2. Introduction part is a little too long and I would like to suggest some to be moved to the discussion

3. Results. Participants were largely female and the authors should discuss the potential reasons for this.

4. Discussion One might argue that lower life satisfaction is either cause or the effect of lower physical activity and the authors should discuss this issue.

5. The authors should discuss more on why elderly people were likely to have less physical activity. Is it because of fear of contracting COVID-19, while it affects more towards elderly with higher mortality? In addition, ff these elderly people start to have more physical activity at places such as gyms, could this potentially increase the risk of infection with higher mortality? The authors need better balanced discussion regarding this issue.

Author Response

We want to thank the reviewers for valuable comments and input, that helped us to further clarify and improve the manuscript. Please see our detailed point-by-point responses and actions below.

Reviewer 2:

  1. Reviewer 2 comment: The abstract should be better informative.

Author’s action: Due to the limitation of 200 words, we are unfortunately limited in adding further information (without removing some other information). We have however added some more numerical results to the abstract (Odds ratio for the 70+ group as well as percentage reporting increase and decrease in PA), which hopefully improves the information.

  1. Reviewer 2 comment: Introduction part is a little too long and I would like to suggest some to be moved to the discussion

Author’s action: We have moved the section regarding physical activity recommendations to the discussion (p.10, row 265-77)

  1. Reviewer 2 comment: Participants were largely female and the authors should discuss the potential reasons for this.

Author’s response: The reason for the over-representation of female responders are likely due to a combination of selection bias from the recruitment via social media, and non-response bias since women are generally more prone to participate in research surveys.
Author’s action: We have added this notion to the methods; p 12, row 3251-57 “Women, and persons with higher education seem to be over-represented in our sample. This may likely be due to a selection bias from the recruitment via social media The current activity levels further indicate that the sample include relatively highly active people This may likely be due to non-response bias leading to an over-representation of persons with interest in physical activity. This may limit the generalizability to the general Swedish adult population, and potentially over-estimate the positive changes in physical activity during the pandemic. However, it is less likely that the between group differences that we found within the sample are affected by this bias.”

  1. Reviewer 2 comment: Discussion One might argue that lower life satisfaction is either cause or the effect of lower physical activity and the authors should discuss this issue.

    Author’s response: Thank you for pointing this out, we have avoided to claim any causal conclusion but we agree that it is an important notion to point out that we can’t draw any firm conclusions regarding direction of causality.
    Author’s action: We have added this notation to the discussion; p 11 row 303-06 “The results were not confounded by factors such as age or current general health status, but due to the cross-sectional design, we can’t draw any conclusion regarding direction of causality. Lower life satisfaction could be both a cause and an effect of decreased activity.”
  2. Reviewer 2 comment: The authors should discuss more on why elderly people were likely to have less physical activity. Is it because of fear of contracting COVID-19, while it affects more towards elderly with higher mortality? In addition, ff these elderly people start to have more physical activity at places such as gyms, could this potentially increase the risk of infection with higher mortality? The authors need better balanced discussion regarding this issue.

Author’s action: Thank you, we have added the following section to the discussion: p 11-12 row 324-34 “It is a delicate balance though, as there are motivated reasons for the sharper recommended restrictions for this age group. Older persons are subject to a higher risk for severe infections and a higher mortality from covid-19 (reference Izkovich et al). Those sharper restrictions seem to have inhibited maintained physical activity in this age group, to a higher extent than in younger age groups that had relatively milder recommended restrictions during the study period. For this older age group though, visiting gyms for example may be associated with a risk for infection, that could be considered to outweigh the potential benefit. Instead, focus should be on encouraging and finding ways to maintain or increase physical activity in forms that are possible to perform while following recommended restrictions regarding e g physical distance. Outdoor exercise or activity, and/or home based exercise programs or online programs that can be followed and performed from home are examples of activities that can be performed with safe physical distance.”

Round 2

Reviewer 1 Report

Dear authors,

Congratulations for the improvements made. You have appropriately justified the issues raised, and did the necessary adjustments in the text (mainly in the methods section: e.g. sensitive analysis testing for controlling factors).

The paper is now ready to be published.

Kind regards.

Reviewer 2 Report

None.